# AMPK and Diseases: State of the Art Regulation by AMPK-Targeting Molecules

**DOI:** 10.3390/biology11071041

**Published:** 2022-07-11

**Authors:** Olga Tarasiuk, Matteo Miceli, Alessandro Di Domizio, Gabriella Nicolini

**Affiliations:** 1Experimental Neurology Unit, School of Medicine and Surgery, University of Milano-Bicocca, 20900 Monza, Italy; gabriella.nicolini@unimib.it; 2SPILLOproject—Innovative In Silico Solutions for Drug R&D and Pharmacology, 20037 Paderno Dugnano, Italy; matteo.asper@gmail.com (M.M.); alessandro.didomizio@spilloproject.com (A.D.D.)

**Keywords:** AMPK regulation, AMP, ADaM site, direct and indirect AMPK activators, AMP mimicking

## Abstract

**Simple Summary:**

AMPK is an enzyme that plays a role in cellular energy mechanisms and controls many metabolic and physiological processes. AMPK is dysregulated in different diseases, making it a potential therapeutic target. AMPK can be activated in a direct or indirect way. In this review, we discuss different AMPK activating compounds and especially focus our attention on those compounds that imitate physiological mechanisms of AMPK activation.

**Abstract:**

5′-adenosine monophosphate (AMP)-activated protein kinase (AMPK) is an enzyme that regulates cellular energy homeostasis, glucose, fatty acid uptake, and oxidation at low cellular ATP levels. AMPK plays an important role in several molecular mechanisms and physiological conditions. It has been shown that AMPK can be dysregulated in different chronic diseases, such as inflammation, diabetes, obesity, and cancer. Due to its fundamental role in physiological and pathological cellular processes, AMPK is considered one of the most important targets for treating different diseases. Over decades, different AMPK targeting compounds have been discovered, starting from those that activate AMPK indirectly by altering intracellular AMP:ATP ratio to compounds that activate AMPK directly by binding to its activation sites. However, indirect altering of intracellular AMP:ATP ratio influences different cellular processes and induces side effects. Direct AMPK activators showed more promising results in eliminating side effects as well as the possibility to engineer drugs for specific AMPK isoforms activation. In this review, we discuss AMPK targeting drugs, especially concentrating on those compounds that activate AMPK by mimicking AMP. These compounds are poorly described in the literature and still, a lot of questions remain unanswered about the exact mechanism of AMP regulation. Future investigation of the mechanism of AMP binding will make it possible to develop new compounds that, in combination with others, can activate AMPK in a synergistic manner.

## 1. AMPK: An Important Player in Regulating Cellular Processes

AMPK is a heterotrimeric complex composed of one catalytic α-subunit and two regulatory β- and γ-subunits. AMPK can consist of two isoforms, each of α- and β-subunits, and three isoforms of γ-subunit [1]. In mammals exist up to 12 distinct combinations of AMPK complexes. These complexes are shown to differ depending on tissue-, cell-type, regulation mechanism, and biochemical properties [2,3].

α-subunit (Figure 1 in blue-grey) consists of the N-terminal kinase domain (KD) (Figure 1, in orange) [4] and C terminus that binds the β (Figure 1, in pink) and γ (Figure 1, in light teal green) subunits and comprises important regulatory domains, namely a so-called autoinhibitory domain (AID) (Figure 1, in red), the α-linker (which interacts with the γ-subunit through two regulatory subunit-interacting motifs, RIM motifs—Figure 1, in blue), and a serine/threonine-rich domain (ST loop).

The β-subunit contains two conserved glycogen-binding domains (GBM) (Figure 1, in bright green), involved in glycogen sensing, and a C terminus domain that binds the α and γ-subunits. The β-subunit also includes a site at its N terminus, which supports the targeting of AMPK to cellular membranes [5].

The γ subunit consists of two domains conserved among all species and called Bateman domains [6], which contain two cystathionine β-synthase repeats (CBS) each, as represented in Figure 2. The four CBS repeats function as four universal adenine nucleotide-binding sites and can competitively bind AMP, ADP, or ATP [7]. In mammals, sites 1 and 3 can exchange adenine nucleotides competitively. Site 2 (CBS2) lacks the key aspartate required for nucleotide binding and consequently is non-functional. Site 4 can also bind AMP and ATP but has a higher affinity for AMP [8].

**Figure 1 biology-11-01041-f001:**
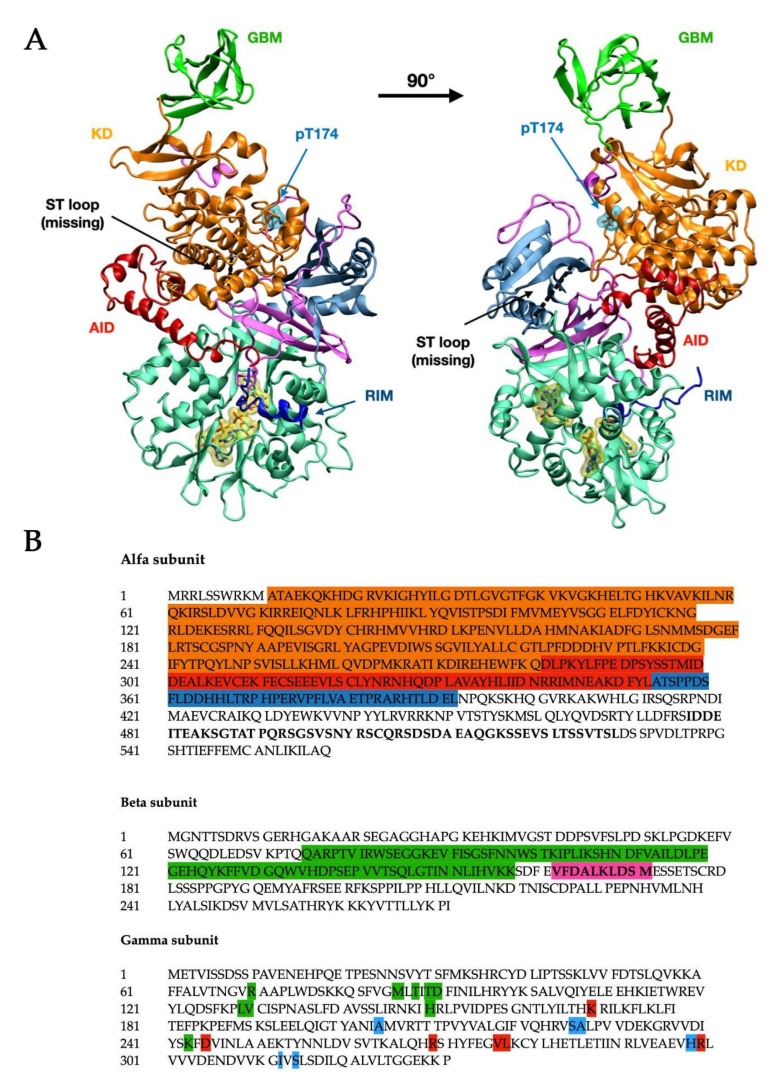
Structural visualization (**A**)and primary sequence (**B**) of human AMPK (PDB ID: 4RER [4]). The structure (**A**) is composed of the α1 subunit (blue-grey) and the β2 (pink) and γ1 (light teal green, with three bound AMP residues) regulatory subunits. The α subunit presents the kinase domain (KD, orange), the autoinhibitory domain (AID, red), and the residue inhibitory motif (RIM, blue). The black dashed line indicates the position of the ST loop, while the phosphorylated Thr-174 (Thr-172 in α2) is shown with the cyan surface. On the β subunit, the glycogen-binding domain (GBM) is shown in bright green. The numbers of starting and ending residues for each highlighted domain are reported in Table 1. In the primary sequence (**B**), the domains are highlighted in the same color code: the α1 subunit (UniProt: Q13131) shows the KD domain (orange), AID (red), and RIM (blue), and the sequence for the ST loop has been bolded. The β2 subunit (UniProt: O43741) shows the GBM domain (green) and the sequence for the ADaM site (bolded, highlighted in pink). In the γ1 subunit (UniProt: P54619), all residues belonging to the respective CBM have been highlighted using the same color code as shown in Figure 2 (green: CBM1, red: CBS3, blue: CBS4).

Important knowledge of AMPK activity regulation by AMP is inferred from the experimentally solved (by X-ray diffraction or cryo-EM) 3D structures of the mammalian AMPK complex. Table 2 reports the 3D structures (including the three α, β, and γ subunits) of human AMPK currently (May 2022) available from the RCSB Protein Data Bank. Recent Cryo-EM structures allow for interesting protein-protein interactions, such as 7JHH.

It is suggested that αRIM, which directly binds AMP at CBS3, starts AMPK activation, making this loop function as an adenine nucleotide sensor. In addition, the N-terminus of the loop folds back and is connected to the AID. When AMP is bound, the C-terminus of the AID and the N-terminus of the αRIM loop form an interaction, termed αRIM-1, that is bound to the unoccupied CBS2 site of the γ-subunit [15].

Direct binding of AMP and/or ADP to γ-subunit allosterically activates the complex and promotes the phosphorylation of Thr-174/Thr-172 (Figure 1, cyan surface) by at least three upstream kinases and three phosphatases: namely, liver kinase B1 (LKB1) [16], calcium-/calmodulin-dependent kinase kinase 2 (CaMKK2) [17], TGFβ-activated kinase 1 (TAK1) [18], protein phosphatase 2A (PP2A) [19], protein phosphatase 2C (PP2C) [20] and Mg^2+^/Mn^2+^-dependent protein phosphatase 1E (PPM1E) [21].

Once AMPK is phosphorylated on Thr-174/Thr-172, AMP preserves AMPK signaling by inhibiting dephosphorylation of pThr-174/172. ATP, on the other hand, competitively inhibits the binding of both AMP and ADP to the γ-subunit, as well as promoting Thr-174/Thr-172 dephosphorylation [22]. The αRIM2 is crucial both for allosteric activation and for protection against dephosphorylation. It is required for sensing the adenine nucleotide binding at γ-subunit and signal transduction to the α-catalytic domain, resulting in either stimulation (AMP) or inhibition (ATP) of AMPK activity [4]. Dephosphorylation of Thr-174/Thr-172 promotes a dramatic conformational change, which brings AMPK from its active (KD associated) to its inactive (KD displaced) state. In this state, the KD is displaced by a distance of around 80 Å from its previous position, thus preventing AMPK catalytic activation [7] (Figure 3).

The ability of the γ subunit to bind AMP, ADP, and ATP allows AMPK to sense the energy state of the cell. Under conditions of high AMP:ATP ratio, AMPK phosphorylates specific enzymes to generate ATP and decrease its consumption; in this way, restoration of energy balance can be reached, acting by turning off energy-consuming processes, such as protein synthesis, and turning on energy-generating processes, mitochondrial biogenesis or glucose metabolism [23]. Cellular ATP concentration is kept at a constant level to ensure adequate ATP supply, which is essential for cellular survival [24].

Due to it, AMPK plays an important role in cell growth and metabolic processes in different tissues such as the liver, muscle, and fat. AMPK directly phosphorylates acetyl-CoA carboxylase 1 (ACC1) and ACC2, suppressing the de novo synthesis of fatty acids and stimulating fatty acids oxidation that regulates overall cellular lipid metabolism [25]. AMPK can induce the inhibitory phosphorylation of the HMG-CoA reductase, which regulates cholesterol synthesis [26]. AMPK stimulates glucose uptake by enhanced GLUT1 and GLUT4 translocation to the plasma membrane and is the main regulator of glucose and fat metabolism in skeletal muscle during exercise [27]. Moreover, AMPK regulates protein synthesis. It is a high energy-consuming process that needs to be inhibited during cellular stress to preserve intracellular ATP. AMPK inhibits cap-dependent translation during both initiation and elongation steps in protein biosynthesis [28]. AMPK downregulates ribosomal RNA synthesis by inducing the inhibitory phosphorylation of transcription initiation factor 1A [29,30]. A recent study suggested that AMPK stimulates cap-independent and IRES-dependent translation of Hif-1α during energy stress to activate the expression of genes important for cell survival [31].

Moreover, different studies have shown that AMPK regulates autophagy processes [32]. First, AMPK directly phosphorylates and activates ULK1 to induce autophagy. Second, AMPK indirectly activates ULK1 by inhibiting mTORC1, which phosphorylates and inhibits ULK1 to disrupt the ULK1–AMPK interaction [33]. This double regulation of ULK1 and mTORC1 eliminates damaged mitochondria and maintains mitochondrial integrity during nutrient starvation [34].

It is also suggested that AMPK regulates mitochondrial homeostasis. AMPK upregulates several antioxidant genes, such as those encoding superoxide dismutase and uncoupling protein 2, which reduces superoxide levels and thioredoxin (TRX), a disulfide reductase, by phosphorylating and activating FOXO, so playing an important role in antioxidant defense during oxidative stress [35]. Some studies suggested that AMPK targets NRF2 and induces antioxidant defense [36].

Considering AMPK’s central role in regulating important cellular metabolic pathways, it has been proposed that it may have therapeutic importance for treating obesity, insulin resistance, type 2 diabetes (T2D), non-alcoholic fatty liver disease (NAFLD), and cardiovascular disease (CVD) [36]. Recently, AMPK has been gaining enormous attention as a target in Alzheimer’s Disease drug research. AMPK activators seem to be able to influence Aβ accumulation, tau aggregation, and oxidative stress [37]. Moreover, strong interest is given to targeting AMPK in cancer. It has been shown that AMPK activation can inhibit tumor cell proliferation and cell growth [38]. In 2003, the discovery of the tumor suppressor LKB1 as the major upstream kinase of AMPK established a link between an energy regulator and cancer pathogenesis, suggesting that the tumor suppressor functions of LKB1 could be mediated by AMPK [39]. Among other mechanisms, several studies showed a correlation between AMPK activation by numerous agents and COX-2 inhibition in different cancers [40]. Moreover, AMPK has been shown to induce phosphorylation of p53 in response to metabolic stress, which is required to initiate AMPK-dependent cell-cycle arrest [41]. Acetyl-CoA carboxylase (ACC) is a well-established downstream target of AMPK involved in lipid metabolism. In several cases, cancer cell proliferation and survival are dependent on ACC activity and inhibiting ACC results in apoptosis [42].

## 2. AMPK Activators

Given that AMPK activation regulates a lot of cellular physiological and pathological processes, it is not surprising that the development of new AMPK activating compounds has been intensively explored. The mechanisms by which compounds activate AMPK can be divided into three classes: (1) compounds that activate AMPK indirectly by increasing intracellular AMP and ADP levels; (2) compounds that activate AMPK directly by selectively binding to its domains; (3) compounds that mimic AMP or ATP and are thus able to bind AMPK at the γ-subunit.

### 2.1. Indirect AMPK Activators

Indirect AMPK activation can be caused by the intracellular accumulation of calcium or AMP. Because AMPK activity is regulated by phosphorylation and dephosphorylation events, the relationships between calcium and upstream kinases or phosphatases play a crucial role. For example, in muscle cells, the increase in cytosolic calcium affects AMPK activation and further influences GLUT-4 gene expression, a skeletal muscle-specific glucose transporter that mediates both insulin and contraction-stimulated glucose transport [43].

AMPK can be activated by compounds that inhibit ATP synthesis as depletion of ATP always causes an increase in AMP and ADP. In cells that are primarily using glycolysis to generate ATP, AMPK is activated by inhibitors of glycolysis such as 2-deoxyglucose. A much larger class of activators is those that inhibit mitochondrial ATP synthesis by inhibiting the respiratory chain, such as metformin, phenformin, antimycin A, oligomycin, and resveratrol [44,45,46]. These agents increase cellular AMP:ATP and ADP:ATP ratios. It is, however, obvious that compounds that inhibit mitochondrial function inhibit oxygen uptake, while those that inhibit glycolysis reduce lactate output and cause extracellular acidification [47]. Due to this, it is expected that different indirect agents, that act mostly by increasing the concentration of AMP, may lack specificity and trigger different unwanted side effects. For example, it has been shown that inhibition of the respiratory chain induced by metformin and phenformin develops life-threatening cases of lactic acidosis that results in phenformin being withdrawn from clinical use [48,49]. Orally administered metformin is effectively absorbed from the gastrointestinal tract to the portal vein. As a first-pass route, the liver is exposed to a high concentration of the drug, which causes gastrointestinal side effects (diarrhea, nausea, abdominal discomfort, anorexia), limiting its use in many patients [47].

There are many other compounds that activate AMPK by inhibiting mitochondrial ATP synthesis, for example, resveratrol, whose pro-oxidative effect leads to cellular oxidative stress limiting its dosage [50]. Moreover, sorafenib can target vascular endothelial growth factor (VEGF) [51], leading to undesirable cardiovascular events like hypertension, bleeding as well as other gastrointestinal disturbances and hand–foot skin reactions [4].

### 2.2. Direct AMPK Activators

Because of indirect AMPK activation side effects, particular attention has been focused on compounds that activate AMPK directly. It is assumed that direct activation of AMPK does not change the AMP:ATP ratio or alter oxygen uptake and does not inhibit mitochondrial function. Direct AMPK activating compounds can be distinguished into two groups: AMP mimetics, which mimic AMP and activate AMPK similarly to physiological ligands, or non-nucleoside activators that bind AMPK at some other sites.

It is shown that some small-molecule activators can bind the allosteric drug and metabolite (ADaM) site between the KD in α-subunit and CBM in β-subunit [24]. ADaM site agonists can activate AMPK both by direct allosteric kinase activation or by protection from dephosphorylation. Adenine nucleotides are strongly present in the cells and are involved in many regulative processes and therefore are poorly selective as AMPK activators [52]. On the contrary, ADaM site activators can be more selective for AMPK and are promising as therapeutic drugs. However, their ineffective activation of β2-subunit, which is the predominant AMPK isoform in the human liver and skeletal muscle is a limitation of many ADaM site agonists [52].

A-769662 was the first molecule for AMPK direct activation developed by Abbott Laboratories in 2006. The development of this direct AMPK activator demonstrated that AMPK activation with non-nucleotide ligands is possible and stimulated to study new approaches for AMPK activation. A-769662 allosterically activates AMPK on the α subunit without Thr-172 phosphorylation [53]. Other direct AMPK activators, compound 991 [14] and MT 63–78 [54], are reported to be 5–10-fold more potent than A-769662. Furthermore, another compound, salicylate, shows structural similarity with A-769662 and binds at a site that overlaps with the A-769662 targeting site. Acetyl salicylate (aspirin) is a derivative that is easier to take orally than salicylate and rapidly breaks down to salicylate upon entering circulation [55]. All four compounds bind the ADaM site and exhibit specificity toward AMPKs that comprise β1 isoforms rather than the β2, giving the possibility to develop isoform-specific AMPK activators.

Discovering new direct AMPK activators appeared very promising in the reduction of lactic acidosis side effects. Vincent et al. compared 6 typical direct (salicylate, A-769662) and indirect (metformin, phenformin, AICAR, 2DG) AMPK agonists that affect cell proliferation. AMPK agonists showed contrasting effects on glycolytic metabolism. Metformin and phenformin increased both glucose consumption and lactate production, while AICAR and 2DG treatment reduced it. On the contrary, salicylate and A-769662 showed no significant changes. These results indicate that indirect AMPK activation may cause unfavorable side effects that can be avoided by direct AMPK activation [56].

Direct AMPK activators showed promising results in eliminating side effects observed from indirect AMPK activation. However, of note are few reports indicating that A-769662 can interfere with various biological pathways unrelated to AMPK through multiple off-target effects [57,58]. For example, it has been shown that A-769662 inhibits the function of the 26S proteasome [57] and Na^+^/K^+^-ATPase [58] activity by an AMPK-independent mechanism. This is calling into question the use of A-769662 as a specific AMPK activator. PT-1 is another compound that directly activates AMPK and promising results have shown increasing AMPK phosphorylation [59]. Unfortunately, besides binding directly to the AMPK α1 subunit, it also induces indirect activation of AMPK via inhibition of the mitochondrial respiratory chain complex, which increases cellular AMP:ATP or ADP:ATP ratios [60].

Another potent, direct, allosteric AMPK activator, MK-8722, can activate 12 mammalian AMPK complexes, mediating AMPK activation in skeletal muscle and inducing robust, durable, insulin-independent glucose uptake and glycogen synthesis, leading to chronically sustainable improvements in glucose homeostasis [61]. However, it has also been noted that it can induce reversible cardiac hypertrophy and increase cardiac glycogen [62]. Although increased cardiac glycogen content and hypertrophy were observed without any changes in electro-cardiogram and apparent functional cardiac sequelae, any safety issues associated with AMPK activators remain to be determined.

### 2.3. AMP Mimicking AMPK Activators

Direct AMPK activators are a good step forward compared to indirect activators; however, their safety aspects need to be considered. Due to this, particular attention is given to compounds that interact with AMPK in its AMP binding site, imitating physiological mechanisms of AMPK activation. Small molecules can mimic cellular AMP and trigger a conformational change and further activation of AMPK by Thr-172 phosphorylation at the α subunit without any significant change in cellular ATP, ADP, or AMP levels [63,64]. The binding of AMP to AMPK can regulate three processes in a synergistic manner: a) promotion of Thr-172 phosphorylation; b) protection against Thr-172 dephosphorylation, and c) allosteric activation of the phosphorylated kinase, making the final response very sensitive to even small changes in AMP [65]. However, very little literature can be found that discusses AMP mimicking compounds and the mechanisms of AMPK activation.

5-aminoimidazole-4-carboxamide riboside (AICAR) is considered to be an AMP mimetic, an adenosine analog taken up by the cell and phosphorylated by adenosine kinase to become ZMP (5-aminoimidazole-4-carboxamide ribonucleotide), which then mimics the activating effect of AMP on AMPK (without affecting the intracellular AMP:ATP ratio) [66]. ZMP is a competitive binder of the natural metabolite AMP and binds to the same CBS domains in AMPK without altering oxygen uptake and inhibiting mitochondrial function, like other AMPK activators. ZMP can accumulate to millimolar concentrations in cells and directly activates AMPK [66]. Even a long-term treatment with AICAR has shown to be beneficial; it enhances glucose tolerance, improves the lipid profile, and reduces systolic blood pressure in an insulin-resistant animal model [67].

Another AMPK activating compound, Cordycepin (30-deoxyadenosine), was reported to be able to enter cells via adenosine transporters. Cordycepin is a prodrug that activates AMPK by being converted by cellular metabolism into the AMP analog cordycepin monophosphate (CoMP) and, in cell-free assays, mimics all three effects of AMP on AMPK [68].

The exciting possibility of a safe ADP mimetic AMPK activator is encouraged by the recently reported by Steneberg et al. [69] successful completion of phase I trials for the novel small molecule O304. O304 is considered to be an ADP mimetic that protects pAMPK against dephosphorylation by the same mechanism as ADP without reducing cellular ATP. However, the exact site of O304 binding has not been reported yet. It is shown that O304, like AMPK activators PF-793 and MK-8722, increases glucose uptake in ex vivo skeletal myotubes in an AMPK-dependent but insulin-independent manner. In mice, O304 increases pAMPK levels and stimulates glucose uptake in skeletal muscle and potently reduces hyperglycemia, hyperinsulinemia, and insulin resistance without inducing cardiac hypertrophy [69].

Activator-3, a recently described AMP mimetic, is a potent AMPK activator that can significantly enhance glucose consumption, increase lipid profiles, reduce body weight, and negative metabolic impact of a high sucrose diet [70]. It interacts with R70 and R152 of the CBS1 domain of the γ subunit near the AMP binding site. A molecular modeling study showed that Activator-3 and AMP likely share a common AMPK activation mechanism. Moreover, it is shown to display a good pharmacokinetic profile in rat blood plasma with low brain penetration [70].

Screening of a chemical library of 1200 AMP mimetics has identified 5-(5-hydroxyl-isoxazol-3-yl)-furan-2-phosphonic acid, termed Compound-2 (C2), and its prodrug C13, as potent allosteric activators of AMPK [71], that has been reported to be >20-fold more potent than A-769662 [72] and more than two orders of magnitude more potent than AMP. Although the precise C2-binding site has not been identified yet, Hunter and colleagues [73] indicated that C2 competes with AMP for binding the γ subunit without causing any significant change in adenine nucleotide levels. However, some further studies suggest that C2 has a dual activation mechanism; Hunter et al. suggested that C2, unlike AICAR or ZMP, is an α1-selective AMPK activator.

Langedorf et al. have described the X-ray crystal structure of full-length α2β1γ1 isoform co-crystallized with C2 and AMP, where they revealed that two molecules of C2 can bind within γ-subunit. In this study, two distinct drug binding sites on AMPK have been identified, one located at the classic α-kinase domain/β-CBM (the so-called ADaM site, see above and Table 1) and the other one within the solvent-accessible core of the γ-subunit (named γ-pSite-1 and γ-pSite-4, respectively). The discovery of γ-subunit C2-binding sites represents a new unknown direction for drug design. This finding can be important for the development of synergistic activation of unphosphorylated α1-AMPK independent of AMP and upstream phosphorylation events [13].

Identification of different activation sites on AMPK raised the idea of dual AMPK targeting, which has been discussed in some studies demonstrating potential results. It has been shown that AMP mimetic compounds and allosteric activators in combination can produce a synergistic effect on AMPK phosphorylation and catalytic activity. Scott et al. have demonstrated that A-769662 and AMP can activate AMPK in a synergistic way (more than 1000-fold) [74]. C2 can synergize with A-769662 as well to activate dephosphorylated AMPK [13].

Ducommun et al. have analyzed the effect of direct AMPK activator A-769662 and the AMP mimetic AICAR, demonstrating their synergistic action on AMPK activation. It is suggested that ZMP binding to AMPK causes the conformational changes of the AMPK complex that facilitates A-769662 binding at β1 but also to β2 AMPK isoforms, which is not occurring when treated by these compounds alone. These data suggest that AMPK activation by two synergistic molecules is more advantageous and allows one to achieve a better physiological effect [75].

Unfortunately, the argument for dual AMPK activation is still poorly analyzed and described in the literature. This is mostly due to the fact that a lot of questions are still unanswered about the exact mechanism of AMPK regulation and activation by AMP. So it remains important to understand the complete composition of the AMP binding site and its mechanism of AMPK regulation to discover new potential compounds that can be used alone or in combination with other AMPK activators. Table 3 summarizes different characteristics of AMPK activators that can be useful for future investigations.

## 3. Gap of Knowledge for AMP Mimicking Drug Development

Although the various crystal structures obtained over the last few years have yielded considerable insight into the mechanism of AMPK regulation by adenine nucleotides, several questions remain.

As mentioned, AMPK senses adenine nucleotides level by AMP, ADP, and ATP binding to CBS1, CBS3, and CBS4 sites in its γ-subunit. However, it is still quite unknown how nucleotides bind to a single site, which nucleotides occupy each site under physiological conditions, and how binding to these sites interferes with each other. Mutagenesis studies demonstrate that CBS3 and CBS4 are important for AMPK allosteric activation. Mutation at I312D on CBS4 showed to inhibit direct allosteric AMPK activation by AMP [8]. It has been further suggested that CBS4 can sense AMP and is important for stabilizing AMP binding at CBS3. CBS4, therefore, allows AMP exchange at CBS3 at nucleotide concentrations that sense cellular energy conditions [52]. In another study, AMPK crystal structure analysis suggests that stabilization is predominantly due to CBS4 blocking the His298 side chain. The His298 backbone directly binds the phosphate group of AMP at CBS3 and blocks the near key residue Arg299, which forms two hydrogen bonds with the AMP phosphate group and two with the AMP adenine ring at CBS3. In addition, αRIM Glu364 blocks CBS3 Arg70 and Lys170, which forms salt bridges with the phosphate group of AMP at CBS3. The phosphate group of AMP carries two negative charges to interact with the three positively charged residues stabilized by αRIM, CBS4, and the partially positive His298 backbone [52]. Beside CBS repeats, α-RIM motives also play an essential role in AMPK regulation. It has been shown that mutating the key α-RIM1 residues largely inhibited the AMP dependence. Mutational analyses demonstrated that the three regulatory elements in the α-subunit, AID, α-RIM1, and α-RIM2, are indispensable for the allosteric activation of AMPK [15]; however, the full mechanism of this regulation remains to be analyzed. Detailed representation of this domain, as well as that of the ADaM site, in available structures, remains to be resolved, given the flexibility of these molecular regions.

Furthermore, it remains unclear AMP, ADP, and ATP differential effects on AMPK containing different γ1/γ2/γ3 isoforms. Unfortunately, most literature that describes full-length AMPK crystal structures studies mainly the γ1 isoform and explains poorly the exact roles of the three nucleotide-binding sites and their regulation mechanisms. The sequences of the CBS repeats in the γ1, γ2, and γ3 isoforms are highly conserved, but it seems possible that small differences of 1, 3, and 4 sites may influence nucleotides affinities and can be important in regulation. It is notable that γ2 and γ3 subunit isoforms have unrelated to each other N-terminal extensions of approximately 250 and 180 residues, respectively, which are not present in γ1. Although the functions of these N-terminal extensions are still uncertain, it is suggested that the N-terminal extensions may be essential for AMPK targeting at specific subcellular locations [88].

Furthermore, it would be important to understand how AMP binding inhibits Thr-172 dephosphorylation [65]. One of the described mechanism suggests that when AMP is bound to site 3, the γ-subunit create interactions with a few amino acids within the α-linkers α-RIM1 and α-RIM2, which interact with the unoccupied site 2 and the AMP molecule bound at site 3, respectively. The binding of the α-RIM motifs to the γ subunit obstructs the flexibility of the α-linker, resulting in stronger interaction of nucleotide-binding, which as a result protects Thr-172 from dephosphorylation. Interestingly, the same mechanism is expected when ADP binds to site 3, suggesting that in some conditions, ADP might be important for AMPK activation [89]. However, it is believed that AMP is a more sensitive controller of AMPK activity than ADP; some studies show that in certain conditions, ADP is a more important activator than AMP. Namely, Coccimiglio et al., in their publication, validated ADP and AMP binding by applying mathematical models suggesting that in skeletal muscle cells during exercise, AMPK activity dynamics are determined principally by ADP and not AMP [90].

It is also interesting what is the exact mechanism of ADP and AMP competition with ATP for γ subunit binding when the cellular ATP amount is usually much higher. A possible explanation for it can be that the Mg-ATP^2−^ binds with a 10-fold lower affinity compared to free ATP^4−^. Main ATP in the cell is present as the Mg-ATP^2−^ form and AMP, ADP may have to compete only with the free ATP^4−^, rather than with Mg-ATP^2−^ complex [23].

## 4. Conclusions

The role of AMPK in maintaining homeostasis in cells and organs made it to be one of the important therapeutic targets in the treatment of different diseases such as obesity, inflammation, diabetes, and above all, cancer. Targeting AMPK studies have taken into account different generations of drug compounds, starting from those that activate AMPK indirectly to more specific direct AMPK activators that target AMPK at the ADaM site or mimic AMP. While indirect AMPK activators hold promise for the treatment of some diseases, it remains uncertain of the different effects of systemic and chronic AMPK activation. Moreover, the complexity of the different AMPK subunit isoforms combinations in tissues and their regulatory properties remains a challenge. In the last decades, significant progress has been made to understand the molecular mechanisms of AMPK regulation that should help develop more potent and specific drugs. Nowadays, the AMPK regulatory motive, the so-called ADaM site, is the most analyzed and discussed in the literature. Different compounds have been developed that target AMPK in this site, and some show promising results. In this review, however, we wanted to pay attention to the other important AMPK activation sites, namely AMP binding sites, their physiological activation site, and the potential for developing AMPK activation compounds that target exactly these sites. AMP binding sites are less analyzed in the literature and still, many questions remain about their exact regulation. Wide knowledge of both AMP and ADaM site mechanisms may facilitate the design of novel therapeutics as well as will open new opportunities in developing new dual synergistic AMPK activation treatment strategies.

## Figures and Tables

**Figure 2 biology-11-01041-f002:**
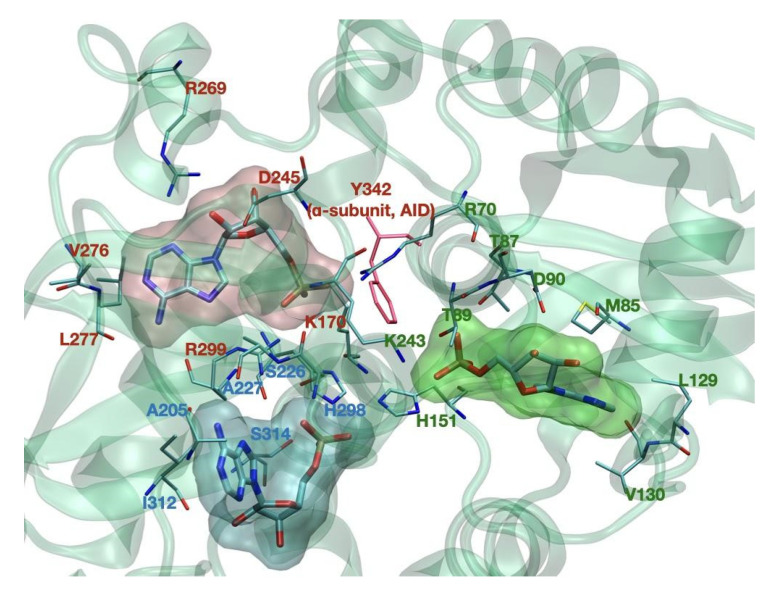
CBS binding sites of the γ subunit. Key binding residues are noted in green for CBS1, in red for CBS3, and in blue for CBS4. Tyr-342, pictured in red, belongs to the AID domain of the α subunit.

**Figure 3 biology-11-01041-f003:**
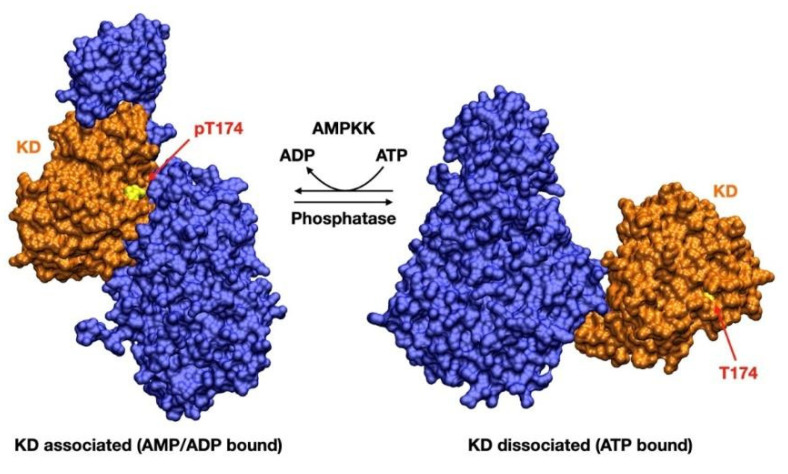
Displacement of the kinase domain (KD) from the associated, active (AMP/ADP bound) conformation. Disassociation is triggered by the substitution of an AMP/ADP residue with Mg-ATP^2−^ one: this brings to a steric clash with the protecting αRIM2 loop, favoring the disassociation of the KD. Phosphorylated Thr-174 (Thr-172 in α2) thus finds itself in an unprotected state, allowing its subsequent dephosphorylation, which in turn inactivates AMPK [4,7,9]. The structures used were 4RER [4] for the associated, active state and 7JHG [9] for the disassociated, inactive state.

**Table 1 biology-11-01041-t001:** Residue numbers for each highlighted domain.

Subunit	Domain	Residues (aa)	Color Code (Figure 1)
Alfa	Kinase Domain	11–281 [4]	Orange
	AID	282–353 [4]	Red
	RIMST loop (black dashed line)	354–392 [4] 477–528 [9]	Blue
Beta	GBM	75–157 [4]	Bright green
	Allosteric drug and metabolite binding site—ADaM site (see below in the text—not completely resolved)	162–171 [10]	n/a

**Table 2 biology-11-01041-t002:** Experimentally determined 3D structures of human AMPK, including the three α, β, and γ subunits, are available (May 2022) from the RCSB Protein Data Bank (https://www.rcsb.org/—accessed on 28 June 2022 ).

PDB ID	Title	Subunits	Method	Release Date	Reference
7JHG	Cryo-EM structure of ATP-bound fully inactive AMPK in complex with Dorsomorphin (Compound C) and Fab-nanobody	α1; β2; γ1	Cryo-EM	2021	[9]
7JHH	Cryo-EM structure of ATP-bound fully inactive AMPK in complex with Fab and nanobody	α1; β2; γ1	Cryo-EM	2021	[9]
7JIJ	ATP-bound AMP-activated protein kinase	α1; β2; γ1	X-ray diffraction	2021	[9]
7M74	ATP-bound AMP-activated protein kinase	α1; β2; γ1	Cryo-EM	2021	[9]
6B2E	Structure of full-length human AMPK (a2b2g1) in complex with a small molecule activator SC4	α2; β2; γ1	X-ray diffraction	2018	[11]
6B1U	Structure of full-length human AMPK (a2b2g1) in complex with a small molecule activator SC4	α2; β1; γ1	X-ray diffraction	2018	[11]
6C9F	AMP-activated protein kinase bound to pharmacological activator R734	α1; β1; γ1	X-ray diffraction	2018	[12]
6C9G	AMP-activated protein kinase bound to pharmacological activator R739	α1; β1; γ1	X-ray diffraction	2018	[12]
6C9H	Non-phosphorylated AMP-activated protein kinase bound to pharmacological activator R734	α1; β1; γ1	X-ray diffraction	2018	[12]
6C9J	AMP-activated protein kinase bound to pharmacological activator R734	α1; β1; γ1	X-ray diffraction	2018	[12]
5ISO	Structure of full-length human AMPK (non-phosphorylated at T-loop) in complex with a small molecule activator, a benzimidazole derivative (991)	α2; β1; γ1	X-ray diffraction	2017	to be published
5EZV	X-ray crystal structure of AMP-activated protein kinase alpha-2/alpha-1 RIM chimaera (alpha-2(1–347)/alpha-1(349–401)/alpha-2(397-end) beta-1 gamma-1) co-crystallized with C2 (5-(5-hydroxyl-isoxazol-3-yl)-furan-2-phosphonic acid)	α2/α1; β1; γ1	X-ray diffraction	2016	[13]
4ZHX	Novel binding site for allosteric activation of AMPK	α2; β1; γ1	X-ray diffraction	2016	[13]
4RER	Crystal structure of the phosphorylated human alpha1 beta2 gamma1 holo-AMPK complex bound to AMP and cyclodextrin	α1; β2; γ1	X-ray diffraction	2014	[4]
4REW	Crystal structure of the non-phosphorylated human alpha1 beta2 gamma1 holo-AMPK complex	α1; β2; γ1	X-ray diffraction	2014	[4]
4CFE	Structure of full-length human AMPK in complex with a small molecule activator, a benzimidazole derivative (991)	α2; β1; γ1	X-ray diffraction	2013	[14]
4CFF	Structure of full-length human AMPK in complex with a small molecule activator, a thienopyridone derivative (A-769662)	α2; β1; γ1	X-ray diffraction	2013	[14]

**Table 3 biology-11-01041-t003:** Summary and characteristics of AMPK activating compounds.

	Mechanism of action	Drug name	Treatment	Disadvantage	Advantage	Ref.
Indirect AMPK activation	intracellular accumulation of Ca^2+^	upstream regulation of (CaMKK2)	calcium-AMPK signaling regulates Human Cytomegalovirus (HCMV) infection	disruption of Ca^2+^ balance can lead to various side effects.	Ca^2+^ plays an essential role in regulating many signaling pathways and cellular processes, such as cell growth and differentiation	[76,77]
	inhibit mitochondrial ATP synthesis by inhibiting the respiratory chain	antimycin A	anti-tumoral	lack of specificity, quite toxic for normal cells	therapeutic advantage for the treatment of tumors	[78]
		metformin	type II diabetes	gastrointestinal side effects (diarrhea, nausea, abdominal discomfort)	glucose-lowering efficacy, modest body weight reduction, easy combination with almost any other glucose-lowering agent	[79]
		phenformin	type II diabetes	develops life-threatening cases of lactic acidosis	potential effect on cancer treatment	[49]
		oligomycin	anti-fungal, anti-tumoral	high lactate accumulating in the blood, urine	therapeutic advantage for the treatment of tumors	[80,81]
		resveratrol	anti-inflammatory, anti-oxidative, antitumoral, neurological, cardiovascular diseases, diabetes, NAFLD, obesity	nausea, vomiting, diarrhea, and liver dysfunction in patients with NAFLD	high range of treatment applications, may promote heart health, lower cholesterol, promote brain health, slow cancer growth	[50]
Direct AMPK activation	ADaM site	A-769662	cardiovascular disorders	can have few off-target effects, ineffective activation of β2-subunit	reduction of lactic acidosis, reduces infarct size, allows a better recovery of contractile function during reperfusion	[82]
	ADaM site	compound 991	skeletal muscle glucose uptake, type II diabetes, obesity	activate β1-isoform 10 times stronger than β2	5-10-fold more potent than A-769662 in activating AMPK, minimal side effects	[83]
	ADaM site	MT 63–78	anti-tumoral	low-affinity binding to β2 subunit	effective at low concentration	[54]
	ADaM site	salicylate	relieve pain and inflammation, reduce fever, prevent excessive blood clotting	difficult breathing, diarrhea, nausea, vomiting	lower risks of cancer, heart disease, and diabetes	[84]
	contradictive information	PT-1	lower hepatic lipid content, type II diabetes, obesity	selective for γ1 and not γ3 isoform	promising AMPK activator, minimal side effects	[59,60]
	ADAM site	MK-8722	increase glucose uptake into skeletal muscle, type II diabetes	induce reversible cardiac hypertrophy and increase cardiac glycogen	activate 12 AMPK complexes, induce robust, durable, insulin-independent glucose uptake and glycogen synthesis	[62]
AMPK mimicking compounds	phosphorylated by adenosine kinase to ZMP, binds the same CBS domains as AMP	AICAR	anti-inflammatory, skeletal muscle glucose uptake, cardiovascular diseases	poor oral bioavailability, may have AMPK-independent effects	long-term treatment without side effects, reduce myocardial infarction	[85]
	AMP analog CoMP binds γ1 subunit	Cordycepin	anti-tumoral, type II diabetes, obesity, anti-fungal, anti-inflammatory, antioxidant, anti-aging, antiviral, hepato-protective	mild gastrointestinal side effects	structure similarity with adenosine makes it an important bioactive component, a wide variety of positive effect	[68,86]
	mimics ADP, suppresses pAMPK. dephosphorylation	O304	type II diabetes, obesity, cardiovascular diseases, peripheral microvascular perfusion	no particular side effects in clinical trials	reduces hyperglycemia, and hyperinsulinemia without inducing cardiac hypertrophy, mimics the beneficial effects of exercise, shows good safety	[87]
	interacts with R70 and R152 of the CBS1 domain on γ-subunit near AMP binding site.	Activator-3	type II diabetes, obesity	mode of activation of Activator-3 is not completely understood	significantly enhance glucose consumption, increase lipid profiles, good pharmacokinetic profile in blood plasma, low brain penetration	[70]
	bind γ -subunit near AMP binding site	Compound-2	metabolic disorders, obesity, and type II diabetes	selectivity for α1 rather than α2 subunit	>20-fold more potent than A-769662 and more than two orders of magnitude more potent than AMP, does not affect any of other AMP activating enzymes	[73]

## Data Availability

Not applicable.

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
