# Peer review of "AMPK and Diseases: State of the Art Regulation by AMPK-Targeting Molecules"

_biology, 2022, doi:10.3390/biology11071041_

Round 1

Reviewer 1 Report

In the article titled "AMPK and diseases: state of the art regulation by AMPK-targeting molecules", the authors summarized different AMPK-targeting compounds. It is well-written and provides useful information for the field of AMPK functions. A minor suggestion: the 3D structure figure in Figure 1 is a little confusing in illustrating the basic domain in AMPK subunits. A linear structure figure may be more clear. And if possible,it would be excellent to provide a diagram depicting the binding site of AMP,ATP and small-molecule activator.

Author Response

Dear Reviewer

We appreciate your precious time in reviewing our paper and providing valuable comments. It was your valuable and insightful comments that led to possible improvements in the current version. The authors have carefully considered the comments and tried our best to address every one of them. We hope the manuscript after careful revisions meet your high standards. The authors welcome further constructive comments if any. Below we provide the point-by-point responses.

Sincerely,

Dr. Olga Tarasiuk

Response to reviewer 1

Comment: A minor suggestion: the 3D structure figure in Figure 1 is a little confusing in illustrating the basic domain in AMPK subunits. A linear structure figure may be more clear. And if possible,it would be excellent to provide a diagram depicting the binding site of AMP,ATP and small-molecule activator.

Response : Figure 1 has been modified and now includes the primary sequence, with all domains noted in the text highlighted using the same color code.

Reviewer 2 Report

Line 34; include cite after γ-subunit.

Line 37; the color description for KD does not match. In fact, on line 59, it says it's orange. Please verify all colors are correct.

Line 43; include cite after available. Please include a citation that supports the inference, or else what is the significance of this paragraph if it is not required for AMPK structural integrity?

Line 45; The order of the acronyms is GBM or CBM (Fig.1, in bright green)?

Line 48; include cite after membranes.

Line 52; include cite after ATP.

Line 64; include cite to support.

Lines 90-94; include citations to support.

Line 106; include cite after metabolism.

Line 112; include cite after metabolism.

Lines 117-120; include citations to support.

Line 125; include cite after processes.

Line 176; include cite after resveratrol.

Line 206; include cite after activators.

Line 219; include cite after circulation.

Line 231; include cite after few reports.

Line 244; include cite after homeostasis.

Line 257; include cite after AMP levels.

Line 267; include cite after ratio.

Line 281; include cite after O304. Please indicate the author(s) of said trial.

Line 191; include cite after diet.

Lines 328-333; There is no evidence for such inferences.

Author Response

Dear Reviewer

We appreciate your precious time in reviewing our paper and providing valuable comments. It was your valuable and insightful comments that led to possible improvements in the current version. The authors have carefully considered the comments and tried our best to address every one of them. We hope the manuscript after careful revisions meet your high standards. The authors welcome further constructive comments if any. Below we provide the point-by-point responses.

Sincerely,

Dr. Olga Tarasiuk

Response to reviewer 2

Comment: Line 34; include cite after γ-subunit.

Response : Citation has been added as suggested

Comment: Line 37; the color description for KD does not match. In fact, on line 59, it says it's orange. Please verify all colors are correct.

Response : Colors were adjusted as suggested

Comment: Line 43; include cite after available. Please include a citation that supports the inference, or else what is the significance of this paragraph if it is not required for AMPK structural integrity?

Response :The paragraph has been removed

Comment: Line 45; The order of the acronyms is GBM or CBM (Fig.1, in bright green)?

Response : Acronym was verified and correspond

Comment: Line 48; include cite after membranes.

Line 52; include cite after ATP.

Line 64; include cite to support.

Lines 90-94; include citations to support.

Line 106; include cite after metabolism.

Line 112; include cite after metabolism.

Lines 117-120; include citations to support.

Line 125; include cite after processes.

Line 176; include cite after resveratrol.

Line 206; include cite after activators.

Line 219; include cite after circulation.

Line 231; include cite after few reports.

Line 244; include cite after homeostasis.

Line 257; include cite after AMP levels.

Line 267; include cite after ratio.

Line 281; include cite after O304. Please indicate the author(s) of said trial.

Line 191; include cite after diet.

Response : All citation were added as suggested

Comment: Lines 328-333; There is no evidence for such inferences.

Response: Dual AMPK activation has been shown to be advantageous; however, there are few publications studying it, namely Ducommun et. al (2014), Scott at.al (2014), Langendorf et.al (2016), Gómez-Escribano et.al. (2020) that are described in the review. With the phrase (line 328-333), we would like to stress out the necessity for more research on dual synergistic AMPK activation.

Reviewer 3 Report

AMPK is an important enzyme that controls the cellular energy homeostasis and a drugging target for treating several chronic metabolic diseases and disorders. In this review, Tarasiuk and coworkers start this review with introducing its composition, structure, molecular mechanisms, cellular functions/regulations, and then focus on discussing the enzyme activators. There is a more comprehensive review published in 2019 (PMID: 30867601) and there seems no significant new results in the field since then. This makes the importance of this work questionable. Anyway, this work can be improved by the following edits.

Major point

It is very important to cite the previous references. I find at least several lines where the author should change/add appropriate references.  For example, line 36, reference is not appropriate; line 48, line 112, line 117, line 126, line 128 , reference should be added for these sentences.   

Minor points

 1. in Table 1, also add a column describing the corresponding color code used in figure 1,for each domain .

2. figure 2, I suggest hiding all the hydrogen atoms, making the ɣ the consistence color as used in figure 1 in transparent surface, deleting the arrows, and zooming in the figure such that the authors can put the residue labels right beside the residue. It would look much better.

3. Table 2. Add a column showing the ligand or activator information for each. Also PDB code acknowledgement is not enough, add another column to cite references for the corresponding publications.

4, Since the focus is the activator, I suggest making a figure to describe the AMPK activation mechanism, to summarize line 91-100.

5. Again, since the focus is the activator, make a table to summarize the whole section 2.

Author Response

Dear Reviewer,

We appreciate you and the reviewers for your precious time in reviewing our paper and providing valuable comments. It was your valuable and insightful comments that led to possible improvements in the current version. The authors have carefully considered the comments and tried our best to address every one of them. We hope the manuscript after careful revisions meet your high standards. The authors welcome further constructive comments if any. Below we provide the point-by-point responses.

Sincerely,

Dr. Olga Tarasiuk

Response to reviewer 3

Comment: It is very important to cite the previous references. I find at least several lines where the author should change/add appropriate references.  For example, line 36, reference is not appropriate; line 48, line 112, line 117, line 126, line 128 , reference should be added for these sentences.  

Response : All citation were added as suggested

Comment: in Table 1, also add a column describing the corresponding color code used in figure 1,for each domain .

Response: The column was added as suggested

Comment: figure 2, I suggest hiding all the hydrogen atoms, making the ɣ the consistent color as used in figure 1 in transparent surface, deleting the arrows, and zooming in the figure such that the authors can put the residue labels right beside the residue. It would look much better.

Response: Figure was modified as suggested

Comment: Table 2. Add a column showing the ligand or activator information for each. Also PDB code acknowledgement is not enough, add another column to cite references for the corresponding publications.

Response: As requested by the reviewer, the references have been added. Regarding the additional information about ligands and/or activators, we appreciate the suggestion; however, this information is already available in the Title of the structure, and we think that adding redundant information would be counterproductive.

Comment: Since the focus is the activator, I suggest making a figure to describe the AMPK activation mechanism, to summarize line 91-100.

Response: The figure was added as suggested

Comment:  Again, since the focus is the activator, make a table to summarize the whole section 2.

Response: The table was added as suggested

Round 2

Reviewer 3 Report

All points are addressed. The authors have done a great job, and  the manuscript is improved significantly.